# Sulfoquinovose Catabolism in *E. coli* Strains: Compositional and Functional Divergence of *yih* Gene Cassettes

**DOI:** 10.3390/ijms262110351

**Published:** 2025-10-24

**Authors:** Anna D. Kaznadzey, Anna A. Rybina, Tatiana A. Bessonova, Dmitriy S. Korshunov, Maria N. Tutukina, Mikhail S. Gelfand

**Affiliations:** 1Center for Molecular and Cellular Biology, Moscow 121205, Russia; rybinaann@gmail.com (A.A.R.); masha306@gmail.com (M.N.T.); mikhail.gelfand@gmail.com (M.S.G.); 2Vavilov Institute for General Genetics of Russian Academy of Sciences, Moscow 117971, Russia; tatianabessonova66@gmail.com; 3Institute of Cell Biophysics of Russian Academy of Sciences, Pushchino 142290, Russia; 4Biology Faculty, Lomonosov Moscow State University, Moscow 119991, Russia; korshunds@yandex.ru

**Keywords:** sulfoquinovose, *Escherichia coli*, Nissle 1917, RNA-seq, bacterial adaptation, *yih* cassette, operon evolution, sulfur metabolism

## Abstract

The sulfo-Embden–Meyerhof–Parnas (sulfo-EMP) pathway enables *Escherichia coli* to utilize sulfoquinovose, (SQ) a sulfonated sugar derived from plant sulfolipids, as a carbon source. This pathway is encoded by the *yih* gene cassette. However, structural and functional diversity of this cassette across *E. coli* strains has not been fully characterized. We identified two structural variants of the *yih* cassette across *E. coli* and *Shigella* strains: a long form (*ompL-yihOPQRSTUVW*) and a truncated short form (*yihTUVW*). Both forms occupy the same genomic location but differ in orientation and are scattered across the phylogenetic tree, suggesting frequent recombination events. Transcriptome analyses revealed that only the long cassette, as found in *E. coli* K-12 MG1655, is transcriptionally induced during growth on sulfoquinovose. The short cassette, found in *E. coli* Nissle 1917 and other host-adapted strains, showed no differential expression. Despite this, both strains grew comparably on sulfoquinovose, indicating different metabolic adaptation strategies. Gene expression profiling revealed shared stress responses but distinct central metabolic patterns. Electrophoretic mobility shift assays further demonstrated that the transcription factor YihW from Nissle 1917 binds its DNA targets with lower affinity than the homolog from K-12 and shows weaker sulfoquinovose-dependent dissociation.

## 1. Introduction

Sulfoquinovose is a sulfonated derivative of glucose that constitutes the hydrophilic head group of sulfoquinovosyl diacylglycerol (SQDG), a major sulfolipid synthesized by photosynthetic organisms such as algae, higher plants, and cyanobacteria [1,2]. An estimated ten billion tons of SQDG are produced annually, representing a significant reservoir of organic sulfur. Bacteria are the main degraders of SQDG and SQ, with six major catabolism pathways currently known [3,4]. These include two glycolytic-like pathways, sulfo-Embden–Meyerhof–Parnas and sulfo-Entner–Doudoroff; two pentose phosphate pathway-like routes, the sulfo-transaldolase and sulfo-transketolase pathways; and two non-glycolytic catabolic pathways, sulfo-ASMO and sulfo-ASDO, which use mono- or dioxygenases to cleave the C–S bond of SQ via alkanesulfonate monooxygenase or dioxygenase, respectively. Among these, the sulfo-EMP pathway is best characterized and serves as a model for SQ degradation [5], particularly in *Escherichia coli* [6].

The sulfo-EMP pathway converts SQ to sulfofructose and then to sulfolactaldehyde and dihydroxyacetone phosphate, the latter entering central glycolysis. Sulfolactaldehyde is further reduced to 2,3-dihydroxypropane-1-sulfonate, completing dissimilation of the sulfonate moiety without desulfonation. A cluster of up to ten genes is responsible for these functions in *E. coli*, referred to as the *yih* cassette. The complete *yih* cassette includes *ompL* (encoding a porin), *yihO* (MFS transporter), *yihP* (MFS transporter), *yihQ* (sulfoquinovosidase), *yihR* (SQ epimerase), *yihS* (SQ isomerase), *yihV* (sulfofructose kinase), *yihT* (sulfofructose-1-phosphate aldolase), *yihU* (sulfolactaldehyde reductase), and *yihW*, alternatively named *csqR* (transcriptional regulator). The respective enzymatic functions allow for conversion of SQ into intermediates of the central carbon metabolism, as described above, ultimately enabling the organism to exploit sulfosugars as a source of carbon and energy. Unlike pathways found in *Pseudomonas putida* and *Paracoccus pantotrophus*, which can mineralize the sulfonate group [5,7], the *E. coli* pathway retains the sulfur in a stable end product (2,3-dihydroxypropane-1-sulfonate), which is subsequently excreted [6].

In total, the sulfo-EMP pathway yields only one pyruvate and one ATP per SQ molecule, providing half the energy and carbon building blocks compared to glycolysis [6]. The NADH/NAD^+^ balance remains unaffected due to the reaction catalyzed by YihU (sulfolactaldehyde reductase). Moreover, the sulfo-EMP pathway does not produce glucose-6-phosphate (G6P) and fructose-6-phosphate (F6P), which are essential for anabolic processes and the pentose phosphate pathway. During growth on SQ as the sole source of carbon, this imposes an additional metabolic burden on the cell, which has to find alternative ways to generate G6P and F6P. In a recent study, the key role of gluconeogenesis in this regard was proposed [8]. The gluconeogenesis pathway compensates for the deficiency of G6P or F6P and might also be accompanied by a shift in metabolic flux away from the tricarboxylic acid cycle.

YihW (CsqR), a transcriptional regulator encoded within the cassette, was identified as a local repressor of the *yih* cassette genes [9,10]. YihW belongs to the DeoR subfamily of GntR-family transcription factors, many of which regulate carbohydrate metabolism in response to small-molecule ligands [11]. Electrophoretic mobility shift assays (EMSA) established that YihW binds to the intergenic regions upstream of its own gene (*yihW*) and of the divergent promoters for *yihV* and *yihU*. SQ inhibits this DNA-binding activity [10], consistent with a model in which SQ acts as an effector that alleviates transcriptional repression in response to substrate availability. In [10], high concentrations of purified YihW protein were needed to bind its DNA targets in the EMSA assays, that possibly reflected the oligomeric nature of this class of regulators, although the precise oligomeric state of YihW remains unresolved.

While the sulfo-EMP pathway is well characterized in the laboratory *E. coli K-12 MG1655* strain, the extent of its variability across *E. coli* lineages and its functional regulation in distinct ecological or pathogenic contexts remain unclear. In particular, little is known about how differences in gene content and organization of the *yih* cassette may influence its transcriptional control and metabolic role. In this study, we compare *E. coli* strains representing distinct cassette architectures to determine how genomic organization affects transcriptional responsiveness to SQ and regulatory binding of the transcription factor YihW (CsqR).

## 2. Results

### 2.1. Long and Short yih Cassettes in E. coli Strains

To assess the variety of the *yih cassettes* across *E. coli* strains and *Shigella* species, we performed homology-based search complemented by gene neighborhood analysis to define cassette boundaries and structural variants. Two distinct types of cassettes were observed, a long variant (*ompL-yihOPQRSTUVW*, first described in *E. coli* K-12 [6] and a short variant (*yihTUVW*), lacking, in particular, genes encoding the initial steps of SQ catabolism, sulfoquinovosidase (*yihQ*), mutarotase (*yihR*), and isomerase (*yihS*). In all analyzed *E. coli* strains carrying the short variant, no homologs of other long cassette genes (*yihO*, *yihP*, *yihQ*, *yihR*, *yihS*) were detected elsewhere in the genome. The short variant is observed in a variety of strains, mostly pathogenic, and, interestingly, in a widely used probiotic strain *E. coli* Nissle 1917 (Figure 1). A significant association was found between the short variant and pathogenic strains (odds ratio = 2.67, *p* = 1.1 × 10^−5^), indicating enrichment in pathogens but not exclusivity. Its presence in *E. coli* Nissle 1917, a member of phylogroup B2 like many extraintestinal pathogenic *E. coli* (ExPEC) strains, may reflect underlying genomic similarity to ExPEC lineages [12].

We analyzed the genomic context of the *yih* cassette in 413 strains using positions of 238 universal single-copy orthologous groups (UOGs) as reference markers with positions highly conserved across most *E. coli* genomes [13]. Specifically, we identified the nearest flanking UOGs upstream and downstream of each *yih* cassette to evaluate the positional consistency across strains and detect possible insertions, deletions, or inversions.

The majority of *yih* cassettes, both short and long, is flanked by genes from UOG217 and UOG236 (Figure 1), indicating that the locus has a conserved genomic position across strains. Notably, the orientation of the short cassette is reversed relative to the long variant.

Analysis of the distribution of cassette types on a phylogenetic tree of *E. coli* and *Shigella* strains (Figure 2) demonstrated that strains with short and long cassettes were interspersed and both types of cassettes were observed in all but one phylogroups (Figure 2). This suggests that the *yih* cassette is an evolutionarily labile module, subject to recurrent rearrangement via homologous recombination or short-distance horizontal gene transfer (HGT) events.

### 2.2. Transcriptional Changes in E. coli Strains with Short and Long yih Cassettes During Growth on Sulfoquinovose

To characterize the transcriptional response of the long and short *yih* cassette genes to SQ, we cultivated *E. coli* K-12 MG1655 and *E. coli* Nissle 1917 in minimal media with SQ or glucose as a sole carbon source to mid-exponential phase and subjected total mRNA to RNA-seq analysis.

We identified differentially expressed genes (DEGs); data obtained with growth on glucose was used as a control (Figure 3). In total, during growth on SQ, *E. coli* K-12 MG1655 had 979 DEGs, of which 236 were upregulated and 743 were downregulated, while *E. coli* Nissle 1917 had 991 DEGs, of which 225 were upregulated and 766 were downregulated. The biological replicates showed consistent expression patterns across the samples (Appendix A).

Among the top DEGs in *E. coli* K-12 MG1655 growing on SQ, we observed genes from the *yih* cassette with a Log2FC (Log2 fold change) five or greater, including transporter genes (*ompL*, *yihO*, and *yihP*), genes encoding sulfo-EMP enzymes isomerase *yihS*, aldolase *yihT*, reductase *yihU*, kinase *yihV,* SQ mutarotase *yihR*, and the transcriptional regulator, *yihW* (*csqR*) (Figure 3, Appendix A). In contrast, *yih* genes of the short cassette in *E. coli* Nissle 1917 did not demonstrate any differential expression in response to SQ (Figure 3, Appendix A).

To validate the RNA-seq results, expression of the *yih cassette* genes was assessed by qRT-PCR after four hours of growth on SQ or glucose for both *E. coli* strains (Figure 4). In *E. coli* K-12 MG1655, the *yihP*, *yihQ*, *yihR*, *yihS*, *yihT*, *yihU*, *yihV*, and *yihW* (*csqR*) genes were significantly upregulated on SQ. No activation of expression was observed for *yihV*, *yihT*, and *yihW* (*csqR)* in *E. coli* Nissle 1917 (Figure 4), while *yihU* was apparently repressed.

Both *E. coli* strains exhibit low growth rates on SQ (Figure 5). As previously reported [6], the biomass of *E. coli* K-12 grown on SQ during mid-log phase was approximately half that of cultures grown on glucose under the same carbohydrate concentrations (OD_600_~0.3 vs. OD_600_~0.6 after four hours of growth) (Figure 5). This can be partially attributed to the fact that the sulfo-EMP pathway generates only half the ATP and carbon yield compared to glycolysis [6].

To assess whether the *E. coli* Nissle 1917 strain used other genes to utilize SQ using, we identified 25 DEGs whose protein products were similar to known bacterial SQ catabolism enzymes. However, these candidates were never co-localized, thus not forming operons or cassettes, and the sequence identity between respective proteins was never higher than 37% suggesting a metabolic function other than utilization of SQ (Appendix A).

Despite transcriptional inactivity of the short *yih* cassette genes, *E. coli* Nissle 1917 maintained growth rates on SQ similar to those of K-12 MG1655, suggesting that it relies on a different catabolic mechanism. Our next objective was to identify similar or contrasting gene expression patterns between the *E. coli* K-12 and Nissle 1917 strains by first identifying putative orthologs using reciprocal best-hit search, followed by gene set enrichment analysis.

### 2.3. Shared and Divergent Gene Expression Patterns in E. coli K-12 and Nissle 1917 in Response to SQ

Both *E. coli* strains grown on sulfoquinovose exhibited transcriptomic signatures consistent with adaptation to nutrient-limited conditions. These included repression of anabolic pathways such as amino acid and vitamin biosynthesis; induction of catabolic programs for alternative carbon sources, notably fatty acids, acetate, and propionate; upregulation of nitrogen-scavenging systems, including arginine and putrescine degradation; and activation of stationary-phase responses, reflected in the expression of genes regulated by RpoS [14] (Figure 6, Appendix A).

More specifically, both strains showed strong upregulation of genes involved in fatty acid (*fadAB*, *fadH*, *fadL*, *fadJI*, *fadD*, *fadM*, *fadE*) and propionate (*prpBCDE*) degradation (Figure 6). Concomitantly, the *aceEF* operon encoding the pyruvate dehydrogenase complex was downregulated.

Nitrogen metabolism was similarly reprogrammed. The *astCADBE* and *ydcSTUV-patD* operons, belonging to the Ntr-Nac regulon, were highly upregulated (Log_2_FC 3–9), indicating reliance on alternative nitrogen sources. Biosynthetic genes for glutamate (*gltBD*), serine (*serA*), aromatic amino acids (*aroL/F*, *tyrA*), and branched-chain amino acids (*ilv*) were downregulated. Sulfur metabolism (*cysPUWAM*, *cysCND*, *cysHIJ*), thiamine biosynthesis (*thiCEFSGH*), and, in *E. coli* Nissle 1917, taurine transport (*tauA*, *tauC*, *tauD*) were suppressed.

Responses of the central carbon metabolism genes differed between the strains.In *E. coli* K-12, genes coding for aconitase (*acnAB*) and fumarase (*fumC)* were upregulated, while malate:quinone oxidoreductase (*mqo*), isocitrate dehydrogenase (*icd*), and dihydrolipoamide dehydrogenase (*lpd*) were downregulated (Appendix A). We also observed downregulation of cytochrome o oxidase (*cyoABCD*) and upregulation of fumarate reductase (*frdABCD*), as well as glycerol catabolism genes (*glpABC*, *glpFKX*), gluconeogenesis marker *glpX* [15], and *treC*.

In contrast, *E. coli* Nissle 1917 featured upregulation of central carbon metabolism genes, including *icd*, and moderate upregulation of *cyoABCDE* (Appendix A). We observed high expression of glyoxylate shunt (*aceAB*, *glcB*) and glycolate metabolism genes (*glcEFGBA*), as well as upregulation of phosphoenolpyruvate carboxykinase (*pckA*), fructose-1,6-bisphosphatase (*fbp*), and phosphoenolpyruvate synthase (*ppsA*).

In K-12, genes of curli biosynthesis (*csgDEFG*) and quorum-sensing regulators (*lsrABCD*, *lsrRK*) were strongly upregulated, (Appendix A). Nissle 1917 showed no differential expression of these genes but demonstrated strong downregulation of flagellar assembly genes (*fliC*, *flgK*, *motAB*, Appendix A).

### 2.4. Binding of YihW (CsqR) to Intergenic Regions of Long and Short yih Cassettes

To determine the binding properties of YihW from Nissle 1917 and its responsiveness to SQ, we performed EMSA with the *yihU/V* and *yihV/W* intergenic regions from *E. coli* K-12 MG1655 (long cassette) and Nissle 1917 (short cassette).

Similarly to the results previously obtained by Shimada et al. [10], YihW from both *E. coli* Nissle 1917 and K-12 MG1655 formed high molecular weight complexes with its DNA targets, possibly due to protein oligomerization, and these complexes dissociated in the presence of SQ acting as effector (Figure 7 and Figure 8). In our experiment, molar excess of the protein required to bind all DNA at 30 °C for the *yihU/V* intergenic region was 24:1 (Figure 7A). At 37 °C, binding was less effective (Figure 7B); even at 48x molar excess of the protein, some DNA remained free. Interestingly, at 30 °C, YihW from Nissle 1917 interacted with *yihU/V* from the parent strain and *E. coli* K-12 MG1655 with equal efficiency (Figure 7A), but at 37 °C, binding to the K-12 fragment was stronger (Figure 7B). YihW (CsqR) from *E. coli* K-12 MG1655 bound all *yihU/V* DNA at 16x molar excess independent of the strain (Figure 7C), suggesting that in Nissle 1917, YihW is less involved in the *yih* cassette regulation compared to K-12, especially at 37 °C.

This assumption is in line with the saturation effect of binding with the *yihV/W* region from the parent strain (Figure 8), suggesting that autoregulatory binding may be inherently weaker or more tightly controlled. It is worth noting that the effect of SQ on complex dissociation between YihW from Nissle 1917 and both intergenic fragments from the parent strain was lower as compared to the fragments from K-12 (Figure 7A and Figure 8).

## 3. Discussion

The *yih* cassette encodes the sulfo-EMP pathway responsible for SQ degradation in *E. coli* K-12 [6]. Here, we report variability in gene content and organization within the cassette across environmental and clinical *E. coli* isolates. The cassette has two major organizational variants: a long form (*ompL-yihOPQRSTUVW*), characteristic of laboratory strains such as *E. coli* K-12 MG1655, and a short form (*yihTUVW*), found in many strains, the majority of which are pathogenic. Interestingly, one of the non-pathogenic strains, containing a short *yih cassette*, is *E. coli* Nissle 1917, widely marketed as a probiotic.

Analyses of phylogeny and synteny (Figure 1 and Figure 2) demonstrate that both variants share a conserved genomic location, flanked by universal orthologous groups, indicating homologous origin. The interspersed distribution of cassette types across phylogenetic clades supports the possibility of horizontal gene transfer and recombination events. The presence of both cassette types in closely related strains further underscores the evolutionary lability of the *yih* cassette and suggests that, in particular, recombination hotspots may facilitate cassette remodeling driven by niche-specific metabolic needs [16].

The short cassettes, which are often found in pathogenic or host-adapted strains, appear in a reversed orientation compared to the long cassettes. This suggests that most likely an inversion and further homologous recombination events have occurred at this locus. Whether short cassettes confer the ability to catabolize SQ remained unclear.

RNA-seq and qRT-PCR analyses (Figure 3 and Figure 4) demonstrated that genes of the long cassette in K-12 MG1655 were transcriptionally responsive to SQ, with significant upregulation of all genes, including those encoding key enzymes of the SQ catabolism (*yihS*, *yihQ, yihR*, *yihT*, *yihV*, and, to lower extent, *yihU*) and associated transporters (*yihO*, *yihP*, *ompL*). The gene of transcriptional regulator *yihW (csqR)* was also upregulated, consistent with substrate-induced derepression. These results align with previous studies that identified YihW as a repressor of the *yih* genes [9,10].

In contrast, genes of the short cassette in *E. coli* Nissle 1917 showed no differential expression during growth on SQ. Neither *yihV*, *yihT*, *yihU*, nor *yihW* were significantly upregulated, and no alternative operons with homologous genes and relevant expression patterns were identified to suggest compensation via a different SQ catabolism pathway. This transcriptional inactivity suggests that the short cassette in *E. coli* Nissle 1917 is no longer involved in SQ degradation. Yet, both *E. coli* K-12 MG1655 and *E. coli* Nissle 1917 showed low, but comparable growth on SQ (Figure 5), suggesting alternative metabolic adaptations in *E. coli* Nissle 1917, possibly relying on gluconeogenesis, fatty acid catabolism, or other pathways. Retention of the long *yih* cassette in environmental strains such as *E. coli* K-12 MG1655, coupled with its active transcriptional response to SQ, supports the hypothesis that these strains exploit plant-derived sulfosugars like SQ, nutrients that are abundant in soil or plant-associated niches.

Gene-set enrichment analysis revealed shared signatures of nutrient stress in both strains during SQ utilization. Biosynthetic genes for amino acids (e.g., *serA*, *gltBD*, *aroL*, *ilv*, *cys*) and cofactors (e.g., *thi* operon) were consistently downregulated, while catabolic pathways for fatty acids (fad genes), propionate (*prpBCDE*), and nitrogen scavenging via arginine and putrescine degradation (*astCADBE*, *ydcSTUV-patD*) were activated. These patterns are consistent with RpoS and Ntr-Nac regulon activation and reflect a shift toward endogenous resource recycling under carbon and nitrogen limitation [14,15]. Suppression of *cys* and taurine transport genes may further indicate sulfur reallocation under stress [17].

Despite these similarities, the strains diverged in their central metabolic responses. *E. coli* K-12 MG1655 showed markers of anaerobic adaptation, gluconeogenesis, and biofilm initiation (e.g., *frdABCD*, *glpX*, *csgDEFG*, *lsr* operon) [14,15], while *E. coli* Nissle 1917 maintained TCA cycle activity, upregulated glyoxylate and glycolate shunts (*aceAB*, *glcEFGBA*), sialic acid degradation genes (*nanK*, *nanE*), and stress-linked membrane proteases [18,19,20,21,22,23,24,25]. Motility was repressed in both strains via distinct routes: upregulation of curli biosynthesis genes (*csgDEFG*) and quorum-sensing regulators (*lsrABCD*, *lsrRK*) in K-12 and downregulation of flagellar genes in Nissle 1917 [21,26].

The regulatory differences were further explored via electrophoretic mobility shift assays, using YihW proteins from both strains (which have the amino acid sequence identity of 72%) and intergenic regions from both cassette types. It was previously demonstrated that YihW (CsqR) from K-12 MG1655 binds to its main target, the *yihU/V* intergenic region, and its own promoter (*yihV/W*) with clear SQ-induced dissociation of complexes [10]. We confirmed this binding pattern, although at much lower protein excess (8×–16× compared to 600×). In contrast, YihW from Nissle 1917 showed significantly weaker binding to its targets at a physiological temperature of 37 °C (Figure 6B for *yihU/V*, and almost no binding for *yihV/W*, Appendix A). Moderate binding was observed at 30 °C—all *yihU/V* DNA was bound at 32× molar excess of YihW (Figure 7), while for *yihV/W* a saturation effect was demonstrated with no difference in binding between 32× and 48× excess of the regulator with free DNA still visible. One possible analogy is the “defensive oversaturation” observed in the *E. coli* GalS system, where high repressor levels lead to autoregulatory binding that buffers against hyper-repression [27]. Notably, the saturation effect was not detected when YihW from *E. coli* Nissle1917 was exposed to the *E. coli* K-12 MG1655-derived DNA fragments (Figure 8). Addition of SQ led to evident dissociation of the YihW-DNA complexes (Figure 7 and Figure 8). However, for both fragments from *E. coli* Nissle 1917, the dissociation effect was weaker than for fragments from *E. coli* K-12 MG1655. Together with the RNA-seq results, this may indicate a less important role of the *yih* cassette for SQ catabolism in this strain. Thus, we suggest that the transcriptional regulator YihW from *E. coli* Nissle 1917 should not be called CsqR (as it is referred to in K-12 MG1655), due to lack of evidence of its involvement in SQ metabolism.

Regulatory rewiring might also explain why the short cassette is retained in probiotic and pathogenic strains despite apparent loss of the SQ catabolism function. In the broader context of transcriptional regulation, it was found that deletion of *yihW* in MG1655 derepresses not only the *yih* cassette genes, but also *ydfG*, an oxidoreductase of unknown relation to the SQ metabolism [10]. This implies that YihW may act as a node in a broader sulfur-responsive regulatory network. This is in line with models of operon evolution where modular gene clusters undergo selective retention, degeneration, or recombination in response to environmental demands [28,29,30,31].

## 4. Materials and Methods

### 4.1. Distribution and Composition of yih Cassettes in E. coli Strains

The genomic and phylogenetic data for *Escherichia coli* and *Shigella* spp. were taken from Seferbekova et al. [13]. The dataset included genomic and coding DNA sequences of *E. coli* and *Shigella* spp., universal single-copy orthologous groups (UOGs). The phylogenetic tree was constructed using a concatenated alignment of 238 universal single-copy orthologous groups with MAFFT (the linsi mode), followed by the Maximum Likelihood inference using RAxML with the GTR+Gamma model and 100 bootstrap replicates.

For the homology search, sequences of the *yih* cassette genes from *E. coli* K-12 MG1655 (GenBank ID: CP032667.1 [32]) were used as queries. Homologs were identified with MMseqs2 v14-7e284 [33], using the mmseqs2 search command with the default parameters against coding DNA sequences of bacterial strains from the considered dataset. The identified homologs of *yih* cassette genes were considered co-localized if the distance between their respective midpoint genomic positions was less than 5000 base pairs.

The phylogenetic tree of *E. coli* strains with resulting *yih* cassettes was visualized using the iTOL web server [34]. To analyze positioning and orientation of the cassettes across different *E. coli* strains, the start coordinate of the first gene and the end coordinate of the last gene in each *yih* cassette was compared with the positions of adjacent UOGs on the bacterial chromosome. Most common compositions of *yih* cassettes flanked by UOGs were visualized using the DNA Features Viewer package version 3.1.3 [35]. To test for an association between the type of *yih* cassettes and pathogenicity, we used metadata on pathotype status from Seferbekova et al. [13]. The distribution of *yih* cassette types among pathogenic and non-pathogenic strains was analyzed using Fisher’s exact test.

### 4.2. Strains and Growth Conditions

For transcriptomics studies, two strains of *E. coli* were used, *E. coli* K-12 MG1655 (Wild type F- lambda- ilvG- rfb-50 rph-1) [36] and *E. coli* Nissle 1917 (Mutaflor^®^, DSM 6601, serotype O6:K5:H1, Herdecke, Germany) [37]. Cell cultures were grown in the minimal medium M9 supplemented with 5% (*v*/*v*) LB and 0.2% (*w*/*v*) of a carbon source, either D-glucose or sulfoquinovose. Bacterial cultures were grown aerobically at 37 °C till mid-exponential phase (OD_600_ = 0.2–0.3) under constant shaking. OD_600_ of each sample was measured using UV–VIS Spectrophotometer (Thermo Scientific, Waltham, MA, USA). Two biological replicates per condition were used for the RNA-seq analysis, and three biological replicates per condition were used for the qRT-PCR validation.

### 4.3. RNA Extraction and qRT-PCR

RNA was extracted using TRIZol (Thermo Fisher Scientific, Waltham, MA, USA) according to the manufacturer’s protocol and then treated with DNAse I (New England Biolabs, Ipswich, MA, USA) for 1 h at 37 °C. Reverse transcription was made using 1 μg of total RNA, gene-specific primers, and MMul-V RevertAid reverse transcriptase (Thermo Fisher Scientific, Vilnius, Lithuania) according to the manufacturer’s protocol. Primers used for reverse transcription (−RT) and amplification (−PCR) are listed in Appendix A. As a control, *hns* was used [37]. The DT-lite thermocycler (DNA-Technology, Moscow, Russia) and qPCR-HS SYBR mix (Evrogen, Moscow, Russia) were used for quantitative PCR (qRT-PCR). Data obtained from three biological samples and no less than three technical replicates were analyzed for statistical significance using the 2^−ΔΔCt^ method. Error bars indicate standard deviations of the respective mean values.

### 4.4. RNA Library Preparation and Sequencing

RNA-seq was performed on two biological replicates per condition. One μg of total RNA from exponentially growing cells was taken for rRNA depletion using the Illumina RiboZero kit (Illumina, San Diego, CA, USA). 100 ng of rRNA-depleted RNA was further taken for library preparation. Sequencing libraries were prepared using the NEBNext Ultra II Directional RNA library prep kit for Illumina (New England Biolabs, Ipswich, MA, USA) ccording to the manufacturer’s protocol and indexed with the NebNext primers (New England Biolabs, Ipswich, MA, USA). Libraries were quantified using the Qubit HS RNA kit (Thermo, Waltham, MA, USA) and qualified on Bioanalyzer (Agilent, Santa Clara, CA, USA). Sequencing was performed as 150 + 150 PE on the Illumina NextSeq 500 machine at the Skoltech Core Genomics and Bioimaging facility. All raw sequencing data have been deposited in the NCBI Sequence Read Archive (SRA) and can be accessed under the BioProject accession number PRJNA1338229.

### 4.5. Analysis of RNA-Sequencing Data

Quality of reads was assessed using FastQC v0.11.9 and MultiQC v1.12.dev0 [38]. Reads were trimmed to remove adapters using BBduk v35.85 [39]. The genome assemblies of *E. coli* K-12 MG1655 (GenBank accession: GCF_000005845.2) and *E. coli* Nissle 1917 (GenBank accession: GCF_019967895.1) were used as references. Reads were mapped to the reference data using Bowtie2 v2.2.1 [40]. Reads were decontaminated of rRNA. All processing of read alignment data was performed using Samtools v1.11 [41]. Read counts were obtained using featureCounts v2.0.1 [42].

Differential expression analysis was performed on two biological replicates for each growth condition (growth on either SQ or glucose) using DESeq2 v1.38.3, with growth on glucose as a baseline condition [43]. Genes with an adjusted *p*-value of <0.05, calculated using the Benjamini–Hochberg method, were selected for further analysis. For visualization purposes, adjusted *p*-values smaller than 1 × 10^−100^ were set to a minimum value of 1 × 10^−100^ to improve readability in the volcano plot and avoid compression of moderately significant data points. All raw values, including uncapped adjusted *p*-values, are available in Appendix A. To compare differential expression data between the two strains, homologs of *E. coli* Nissle 1917 genes were identified in the *E. coli* K-12 MG1655 genome by reciprocal best-hit search with MMseqs2 v13-45111 [43]. Functional analysis of differentially expressed genes (DEGs) was performed using the clusterProfiler v4.6.2 [44], pathview v1.38.0 [45], KEGGREST, v1.38.0 and DOSE v3.24.2 [46] R packages as follows: To determine which pathways respond most prominently to growth on SQ, DEGs from each strain were mapped to the KEGG genome data of *E. coli* K-12 MG1655 and gene set enrichment analysis was conducted on DEGs within specific KEGG pathways (*p*-value < 0.05, the Benjamini–Hochberg adjusted *p*-value method). Functional annotation of DEGs was cross-validated using the DAVID web-server [47]. Operons containing the DEGs in the reference genome of each strain were predicted using Operon-mapper [48].

### 4.6. YihW Expression and Purification

Both protein variants were expressed under control of T7 promoter and own Shine-Dalgarno and regulatory regions in the pGEMEX-based vectors [49,50]. YihW from *E. coli* K-12 MG1655 was produced as described in Rybina et al. [49] and purified using the HIStrap column (GE Healthcare, Chicago, IL, USA) according to the manufacturer’s protocol. YihW from Nissle 1917 was produced in BL21(DE3)-Codon Plus-RIL cells, induced with 50 μM IPTG at OD600 = 0.2 and incubated for 5 h at 37 °C in LB (Luria–Bertani) media. Then cells from 200 mL of culture were harvested, washed with 1× BS (137 mM NaCl, 2.7 mM KCl, 10 mM Na2HPO4, 1.8 mM KH2PO4, pH 7.4), resuspended in 1× PBS containing 1 mM DTT and 1 mM PMSF, and sonicated on the Q125 Sonicator (QSonica, LLC, Newtown, CT, USA) for 15 min (10 s pulse, 20 s pause) at 45% intensity. Cleared lysate was removed, the pellet was again washed, resuspended in 2 mL of Loading buffer (50 mM NaH2PO4, 300 mM NaCl, 5 mM imidazole, 8 M urea), and then the protein was purified on the Ni-NTA spin columns (Qiagen, Hilden, Germany) using modified buffers containing both urea and imidazole (5 mM for equilibration, 20 mM for washes, 500 mM for elution). Purified protein was then dialyzed against buffers containing 50 mM NaH2PO4 and 300 mM NaCl with lowering urea concentrations and used for band-shift assays (Appendix A).

### 4.7. Electrophoretic Mobility Shift Assays

Electrophoretic mobility shift assays (EMSA, band-shift assays) were used to test the efficiency of the YihW proteins binding to the regulatory regions of genes from the *yih* cassette. DNA fragments containing intergenic regions *yihUV* and *yihVW* were PCR amplified (primers marked as R and F for each intergenic region, Appendix A), extracted from the gel and used for subsequent experiments. 0.5 pmol of DNA was incubated at 30 °C or 37 °C in 1× Binding buffer (50 mM Tris-Cl, 100 mM KCl, 50% glycerol, 1mM EDTA) and appropriate amount of SQ [10] synthesized as described in [31] for 10 min, then 2 to 24 pmol of purified YihW was added. Free DNA fragments were loaded on separate lanes as controls. After further 30 min incubation, complexes were loaded onto a 5% polyacrylamide gel that had been pre-warmed to 30 °C or 37 °C, respectively. Gels were run in 1xTBE buffer under 200V, stained with ethidium bromide and visualized on the Vilber gel-documentation system (Collégien, France).

## Figures and Tables

**Figure 1 ijms-26-10351-f001:**
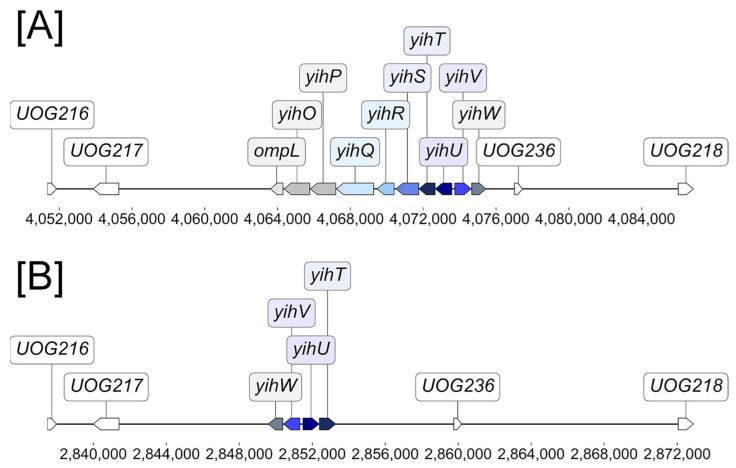
Position of *yih cassettes* relative to UOGs: (**A**) Most common configuration of the long cassette, represented by *E. coli* K-12 MG1655; (**B**) Most common configuration of the short cassette, represented by *E. coli* Nissle 1917.

**Figure 2 ijms-26-10351-f002:**
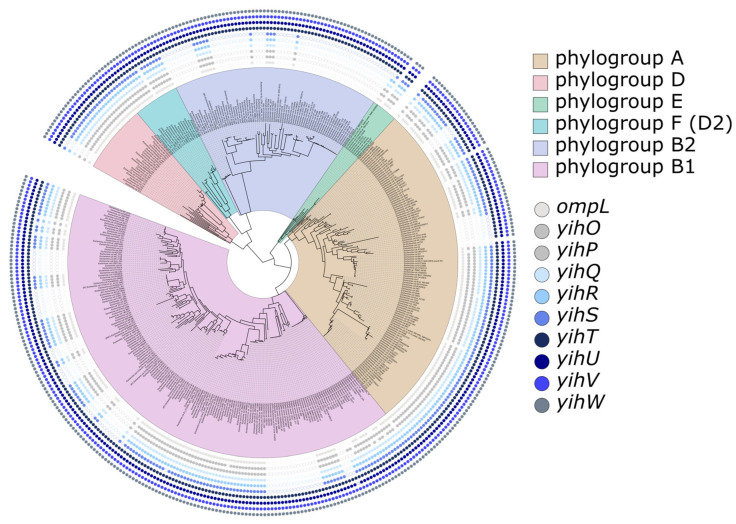
Phylogenetic tree of *Escherichia coli* and *Shigella* spp. Phylogroups are marked by different colors. The presence of *yih* genes is indicated by colored dots, revealing two distinct types: a long variant (e.g., *ompLyihOPQRTSUVW* in *E.coli* K12 MG1655) and a short variant (e.g., *yihTUVW* in *E. coli* Nissle 1917).

**Figure 3 ijms-26-10351-f003:**
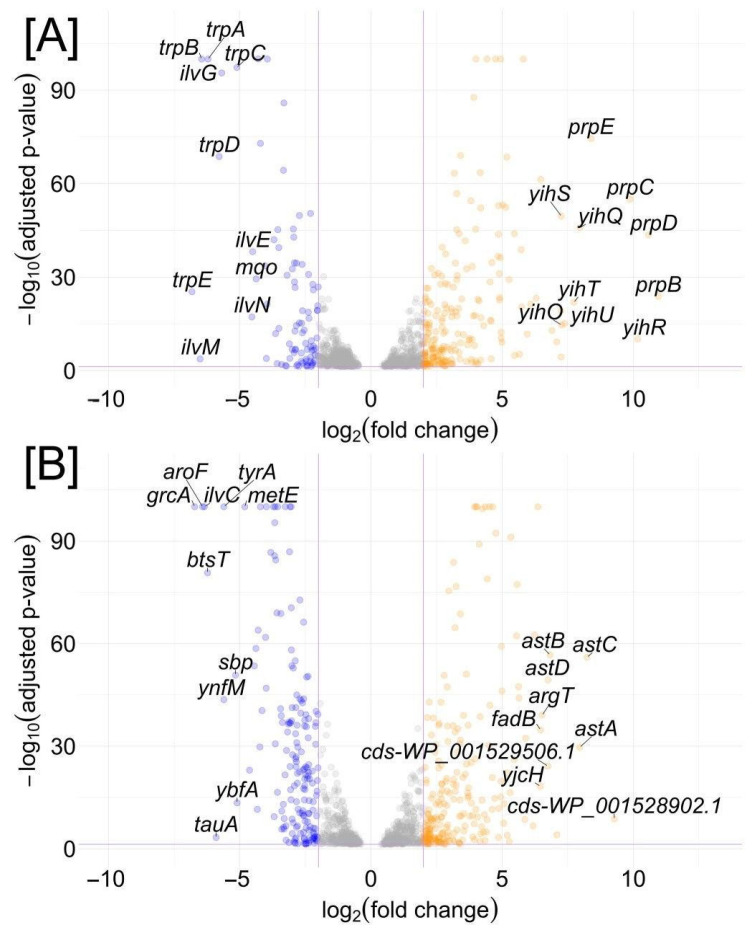
Distribution of DEGs in *E. coli* K-12 (**A**) and *E. coli* Nissle 1917 (**B**) during growth on SQ compared to growth on glucose. Upregulated genes are marked orange, downregulated genes are marked blue, genes without significant changes in expression are marked gray. The top ten DEGs are labeled. The x-axis represents log_2_(fold change), the y-axis represents the −log_10_(adjusted *p*-value). Thresholds for significance are indicated by violet dashed lines at log_2_(fold change) = ±2 and −log_10_(adjusted *p*-value) ≈ 1.3 (corresponding to an adjusted *p*-value of 0.05).

**Figure 4 ijms-26-10351-f004:**
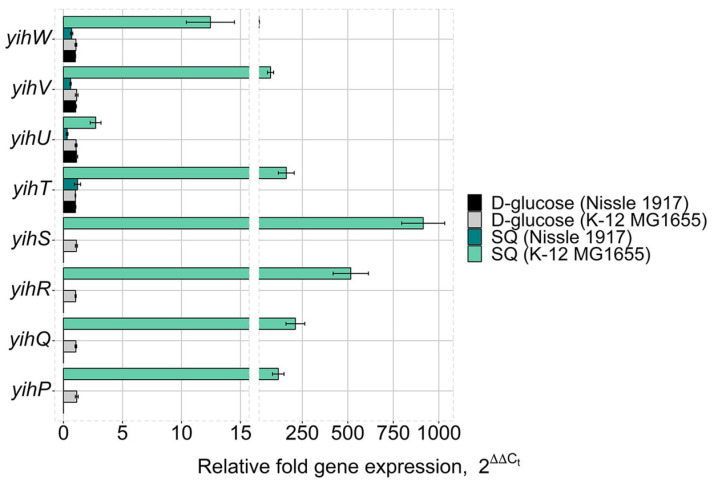
Expression of the *yih* cassette genes in *E. coli* K-12 MG1655 and *E. coli* Nissle 1917 after four hours of growth on D-glucose or SQ as a sole carbon source. Standard deviations were calculated based on three biological and four technical replicates.

**Figure 5 ijms-26-10351-f005:**
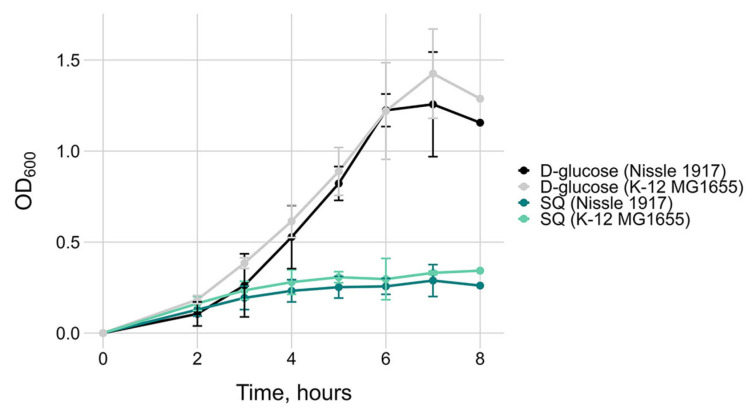
Growth of *E. coli* K-12 MG1655 and *E. coli* Nissle 1917 strains in the presence of 0.2% D-glucose or SQ as sole carbon source. Mean values and 95% confidence intervals were obtained on three biological replicates.

**Figure 6 ijms-26-10351-f006:**
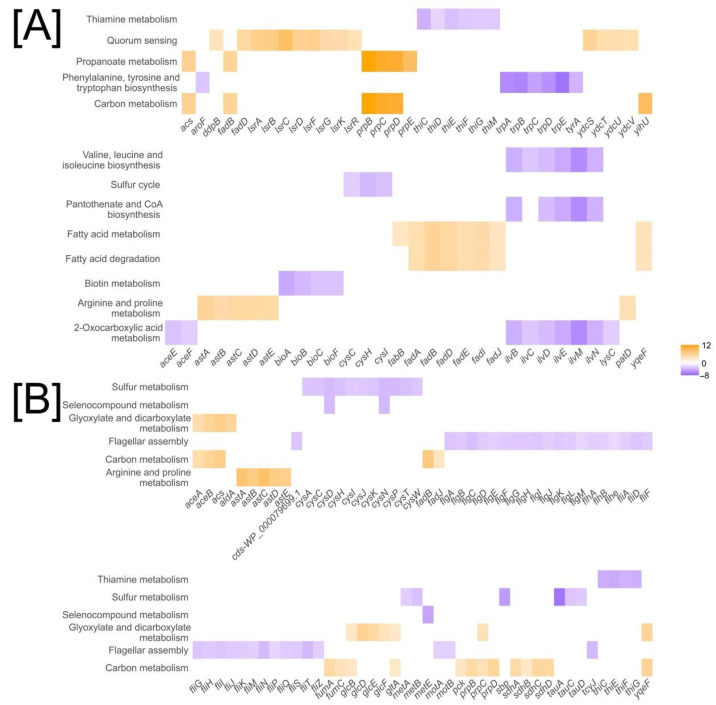
Enriched KEGG pathways in *E. coli* K-12 MG1655 (**A**) and *E. coli* Nissle 1917 (**B**) grown on SQ. Gene categories with the most enriched pathways are colored based on their log_2_(fold change) values, where blue indicates downregulation, orange indicates upregulation, and gray represents mid-range expression levels.

**Figure 7 ijms-26-10351-f007:**
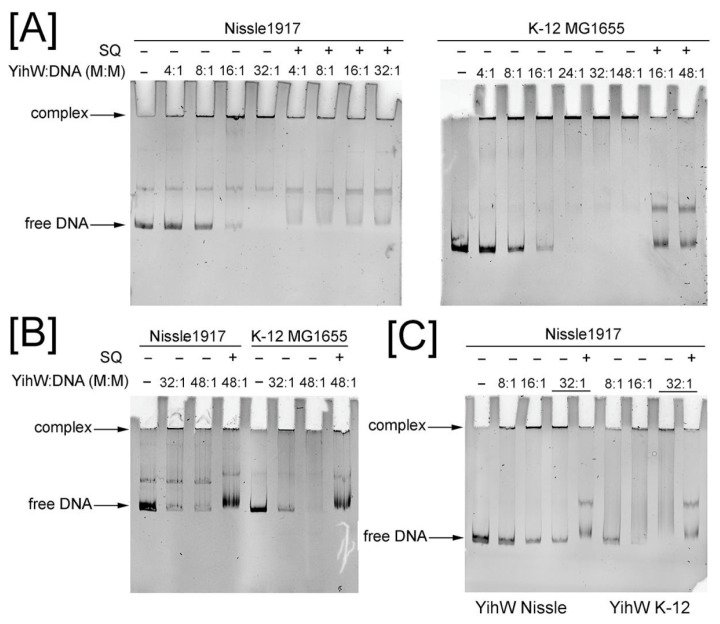
Binding of YihW from *E. coli* Nissle 1917 to the *yihU/V* intergenic region from *E. coli* Nissle 1917 and *E. coli* K-12 MG1655. (**A**) 30 °C (**B**) 37 °C. The YihW:DNA molar ratio, template strains, and the presence of SQ are indicated above the lanes. (**C**) Comparison of binding properties of YihW from *E. coli* Nissle 1917 and YihW (CsqR) from *E. coli* K-12 MG1655 with the *yihU/V* intergenic region from *E. coli* Nissle 1917 at 37 °C. The protein origin is indicated below the lanes.

**Figure 8 ijms-26-10351-f008:**
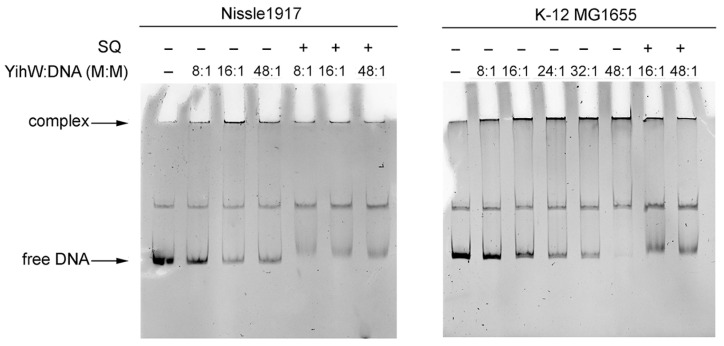
Binding of YihW from *E. coli* Nissle 1917 to the *yihV/W* intergenic region from *E. coli* Nissle1917 and *E. coli* K-12 MG1655 at 30 °C. The YihW:DNA molar ratio, template strains, and the presence of SQ are indicated above the lanes.

## Data Availability

The RNA-seq data generated in this study have been deposited in the NCBI Sequence Read Archive (SRA) under the BioProject accession number PRJNA1338229. The code used for RNA-seq preprocessing, differential expression analysis, and phylogenetic annotation is available at: https://github.com/rybinaanya/for_yih_consideration (accessed on 21 October 2025).

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
