# Peer review of "Sulfoquinovose Catabolism in E. coli Strains: Compositional and Functional Divergence of yih Gene Cassettes"

_ijms, 2025, doi:10.3390/ijms262110351_

Round 1

Reviewer 1 Report

Comments and Suggestions for Authors

Kaznadzey et al. present a valuable and well-structured study on the investigation of yih gene cassettes in E. coli. The manuscript is generally well-written, and the research question is relevant. However, the current version has several substantial issues that need to be carefully addressed before the work can be considered for publication.

Comments: 

1. The Data Availability Statement is absent.

From the author guidelines (https://www.mdpi.com/journal/ijms/instructions#sequence):
New sequence information must be deposited to the appropriate database prior to submission of the manuscript. Accession numbers provided by the database should be included in the submitted manuscript. Manuscripts will not be published until the accession number is provided.

In L. 393-393, the authors report the acquisition of new sequencing data. Thus, I cannot recommend this paper for publication until the authors deposit the reported sequencing data in an appropriate public repository and include the corresponding accession numbers in the manuscript.

2. L. 78-87. In the last paragraph of the Introduction, the authors briefly report their obtained results, which is not appropriate for this section. The Introduction should provide background, identify knowledge gaps, and logically lead to the objectives or hypotheses of the study. Reporting results here disrupts the narrative flow and blurs the distinction between sections. For this reason, I strongly recommend that the authors rewrite this paragraph and clearly state, at its end, the main aims of the study or the central hypothesis being tested. The detailed results should be presented exclusively in the Results section, which is specifically designated for this purpose.

3. L. 346. Here and elsewhere, please consider writing “from Seferbekova et al. study [12]” instead of simply “from [12]”. Using only a reference number interrupts the readability of the text and forces the reader to check the reference list to understand the context. Explicitly naming the authors makes the narrative more fluent, improves clarity, and helps highlight the contribution of the cited work, rather than reducing it to an anonymous number in brackets.

4. L. 373 and 385. Please indicate the number of biological replicates directly in these sentences. Currently, the replicates are only mentioned much later in the corresponding paragraphs, which reduces the clarity and readability of the experimental description. I recommend revising these formulations so that the number of replicates appears upfront, making the methodology transparent and reproducible from the very first mention.

5. L. 402-404. The differential expression analysis uses two biological replicates per condition. This is the bare minimum for DESeq2 and is generally insufficient to obtain reliable dispersion estimates and to detect moderate effects at an acceptable FDR. Please state whether you performed an a priori sample-size/power analysis (sample size justification), including the assumed dispersion, library size, target FDR, expected log2 fold change (log2FC), desired power, and the resulting minimum detectable effect size. For example, the RNASeqPower package (R) can be used for such calculations.

The use of only two biological replicates per condition may partly account for the high number of DEGs reported in L. 125–126.

Additionally, I note that the authors validated their RNA-seq results with RT-PCR, which alleviates concerns about the significance of the findings. Nevertheless, it is surprising that the RNA-seq investigation was designed with only two biological replicates per condition.

6. L. 355-356. The description of the phylogenetic analysis is insufficient. The authors do not specify which inference approach was used (e.g., Maximum Likelihood, Bayesian inference, or distance-based methods), nor do they provide essential technical details such as the substitution model applied, the number of bootstrap replicates (if applicable), or other relevant parameters. The absence of this information undermines the reproducibility and transparency of the presented phylogenetic results.

7. Figure 2. Legend labels have a partial gray background (the fill does not cover all characters). Please either remove the gray highlight entirely or apply a uniform background that fully spans each label so all letters are covered.

8. Figure 6. The current placement of the shared legend for panels 6A and 6B sits between/inside the heatmaps, breaks the reading flow, and creates visual imbalance by competing with the data. Please move the legend to the right or to the bottom of the combined panels and align it with the panel grid and margins.

If the two panels are meant to be compared directly, use a single common color scale and one legend; if not, keep separate, clearly labeled legends (their ranges differ) but place them outside the plotting area. In R, this can be done with patchwork (e.g., plot_layout(guides = "collect") and a guide_area() positioned to the right or bottom).

9. Please consider sharing the code for the implemented data-analysis pipeline to support reproducibility.

Author Response

Reviewer 1

Comments 1. The Data Availability Statement is absent.

From the author guidelines (https://www.mdpi.com/journal/ijms/instructions#sequence):
New sequence information must be deposited to the appropriate database prior to submission of the manuscript. Accession numbers provided by the database should be included in the submitted manuscript. Manuscripts will not be published until the accession number is provided.

In L. 393-393, the authors report the acquisition of new sequencing data. Thus, I cannot recommend this paper for publication until the authors deposit the reported sequencing data in an appropriate public repository and include the corresponding accession numbers in the manuscript.

Response 1: Thank you for the comment. The RNA-seq data generated in this study have been deposited in the NCBI Sequence Read Archive under BioProject accession PRJNA1338229. It is now stated in the “Data Availability Statement” at the end of the manuscript and in the “RNA library preparation and sequencing” subsection of the Methods.

Comments 2. L. 78-87. In the last paragraph of the Introduction, the authors briefly report their obtained results, which is not appropriate for this section. The Introduction should provide background, identify knowledge gaps, and logically lead to the objectives or hypotheses of the study. Reporting results here disrupts the narrative flow and blurs the distinction between sections. For this reason, I strongly recommend that the authors rewrite this paragraph and clearly state, at its end, the main aims of the study or the central hypothesis being tested. The detailed results should be presented exclusively in the Results section, which is specifically designated for this purpose.

Response 2: In the revised version, we have rewritten this paragraph to remove statements that could be interpreted as results. The revised text now focuses on outlining the existing knowledge gap and the main objectives of the study.

Comments 3. L. 346. Here and elsewhere, please consider writing “from Seferbekova et al. study [12]” instead of simply “from [12]”. Using only a reference number interrupts the readability of the text and forces the reader to check the reference list to understand the context. Explicitly naming the authors makes the narrative more fluent, improves clarity, and helps highlight the contribution of the cited work, rather than reducing it to an anonymous number in brackets.

Response 3: Thank you for the suggestion, we have changed the links according to this suggestion.

Comments 4. L. 373 and 385. Please indicate the number of biological replicates directly in these sentences. Currently, the replicates are only mentioned much later in the corresponding paragraphs, which reduces the clarity and readability of the experimental description. I recommend revising these formulations so that the number of replicates appears upfront, making the methodology transparent and reproducible from the very first mention.

Response 4: We have now added information about the number of biological replicates in the relevant Methods sections: “Strains and growth conditions” (“Two biological replicates per condition were used for the RNA-seq analysis, and three biological replicates per condition were used for the qRT-PCR validation.”) and “RNA library preparation and sequencing” (“RNA-seq was performed on two biological replicates per condition.”).

Comments 5. L. 402-404. The differential expression analysis uses two biological replicates per condition. This is the bare minimum for DESeq2 and is generally insufficient to obtain reliable dispersion estimates and to detect moderate effects at an acceptable FDR. Please state whether you performed an a priori sample-size/power analysis (sample size justification), including the assumed dispersion, library size, target FDR, expected log2 fold change (log2FC), desired power, and the resulting minimum detectable effect size. For example, the RNASeqPower package (R) can be used for such calculations.

The use of only two biological replicates per condition may partly account for the high number of DEGs reported in L. 125–126.

Additionally, I note that the authors validated their RNA-seq results with RT-PCR, which alleviates concerns about the significance of the findings. Nevertheless, it is surprising that the RNA-seq investigation was designed with only two biological replicates per condition.

Response 5: We acknowledge that using two biological replicates per condition represents the minimal acceptable design for DESeq2 and may limit statistical power, particularly for detecting moderate expression changes. No formal a priori power analysis was performed. However, our experimental setup adhered to DESeq2’s technical requirements for the variance estimation, and the biological replicates demonstrated high concordance, supporting data reliability. Importantly, as the reviewer noted, key differential expression findings were independently validated by qRT-PCR, which confirmed the direction and magnitude of major expression changes, thereby strengthening confidence in the main conclusions.

Comments 6. L. 355-356. The description of the phylogenetic analysis is insufficient. The authors do not specify which inference approach was used (e.g., Maximum Likelihood, Bayesian inference, or distance-based methods), nor do they provide essential technical details such as the substitution model applied, the number of bootstrap replicates (if applicable), or other relevant parameters. The absence of this information undermines the reproducibility and transparency of the presented phylogenetic results.

Response 6: The phylogenetic tree used in our study was obtained from Seferbekova et al. (2021) [Ref. 12], who constructed it using a concatenated alignment of 238 universal single-copy orthologs. The alignments were generated using MAFFT (linsi), and the tree was inferred using RAxML with the GTR+Gamma substitution model and 100 bootstrap replicates. We have now added this information to the Methods section (“Distribution and composition of yih cassettes in E. coli strains”): “The tree was constructed using a concatenated alignment of 238 universal single-copy orthologous groups with MAFFT (the linsi mode), followed by the Maximum Likelihood inference using RAxML with the GTR+Gamma model and 100 bootstrap replicates [12].”

Comments 7. Figure 2. Legend labels have a partial gray background (the fill does not cover all characters). Please either remove the gray highlight entirely or apply a uniform background that fully spans each label so all letters are covered.

Response 7: We thank the reviewer for noticing this issue. The gray background in the legend labels of Figure 2 has been removed entirely for a cleaner and more consistent appearance.

Comments 8. Figure 6. The current placement of the shared legend for panels 6A and 6B sits between/inside the heatmaps, breaks the reading flow, and creates visual imbalance by competing with the data. Please move the legend to the right or to the bottom of the combined panels and align it with the panel grid and margins.

If the two panels are meant to be compared directly, use a single common color scale and one legend; if not, keep separate, clearly labeled legends (their ranges differ) but place them outside the plotting area. In R, this can be done with patchwork (e.g., plot_layout(guides = "collect") and a guide_area() positioned to the right or bottom).

Response 8: Figure 6 has been revised and the single color scale legend has been repositioned outside the plotting area and aligned with the panel grid for improved readability and visual balance.

Comments 9. Please consider sharing the code for the implemented data-analysis pipeline to support reproducibility.

Response 9: We have created a publicly available GitHub repository containing the scripts used in this study, including RNA-seq preprocessing, differential expression analysis, and phylogenetic annotation steps. The repository is accessible at: https://github.com/rybinaanya/for_yih_consideration. This link has been added to the Data Availability Statement in the manuscript.

Reviewer 2 Report

Comments and Suggestions for Authors

Sulfoquinovose (6-deoxy-6-sulfo-d-glucose, SQ) is the polar headgroup of sulfolipids widely present in plants and one of the most abundant organosulfur compounds. Thus, degradation of SQ is an important part of the global sulfur cycle. Update, several pathways to degrade SQ have been discovered in bacteria, of which sulfo-EMP pathway is mainly reported in Gram-negative bacteria. The sulfo-EMP is composited by the Yih cassette in Escherichia coli. This manuscript reported two structural variants of the yih cassette across in E. coli, a long form (ompL-yihOPQRSTUVW) and a truncated short form (yihTUVW). Further, comparative transcriptomics was applied to characterization of the transcriptional response of two E. coli strains (hosting a long form and a short form of yih cassette, respectively) to SQ. The data showed transcripts of the long form of yih cassette was inducible in response to SQ, but not for the short form. Finally, the repressor yihW from the two different E. coli strains showed distinct affinity to the target DNA. These results imply there are different mechanisms to degrade and utilize SQ by various E. coli strains.

In the introduction, the authors are encouraged to describe the other pathways to degrade SQ in bacteria briefly, except the sulfo-EMP pathway.

The authors found a short form (yihTUVW) of the yih cassette in some E. coli strains, in comparison with the full form (yihOPQRSTUVW). Was the deleted part such as yihOPRQRS found at the other location or dispensed in the genome of these strains like Nissle 1917? Did the authors search the genomes and find the other SQ degradation pathways in these trains hosting the short form of yih cassette?

In EMSA experiment, the purified recombinant proteins of YihW derived from both E. coli strains should be presented as in the from of SDS-PAGE. Further, the YihW proteins were purified through denaturing/refolding. Did you obtain the soluble or active form actually? The protein/DNA complex is somehow unusual in the native gels, which did not perform mobility. Besides, two DNA bands were observed in the EMSA gel. Anyway, the EMSA experiments results should be further confirmed.

The binding assay was performed at two temperatures. Did these two E. coli strains show different response to SQ at various temperatures?

Finally, reference citation in the text should be kept in the order and some supplementary files such as Fig. S2 and Fig. S4 were not cited in the text. The RNA-seq data should be provided availably.

Comments on the Quality of English Language

N.A.

Author Response

Comments 1: Sulfoquinovose (6-deoxy-6-sulfo-d-glucose, SQ) is the polar headgroup of sulfolipids widely present in plants and one of the most abundant organosulfur compounds. Thus, degradation of SQ is an important part of the global sulfur cycle. Update, several pathways to degrade SQ have been discovered in bacteria, of which sulfo-EMP pathway is mainly reported in Gram-negative bacteria. The sulfo-EMP is composited by the Yih cassette in Escherichia coli. This manuscript reported two structural variants of the yih cassette across in E. coli, a long form (ompL-yihOPQRSTUVW) and a truncated short form (yihTUVW). Further, comparative transcriptomics was applied to characterization of the transcriptional response of two E. coli strains (hosting a long form and a short form of yih cassette, respectively) to SQ. The data showed transcripts of the long form of yih cassette was inducible in response to SQ, but not for the short form. Finally, the repressor yihW from the two different E. coli strains showed distinct affinity to the target DNA. These results imply there are different mechanisms to degrade and utilize SQ by various E. coli strains.

In the introduction, the authors are encouraged to describe the other pathways to degrade SQ in bacteria briefly, except the sulfo-EMP pathway.

Response 1: We agree with this suggestion. We have revised the Introduction to describe all six known bacterial pathways for sulfoquinovose degradation: “Bacteria are the main degraders of SQDG and SQ, with six major catabolism pathways currently known [3,4]. These include two glycolytic-like pathways, sulfo-Embden–Meyerhof–Parnas and sulfo-Entner–Doudoroff; two pentose phosphate pathway-like routes, the sulfo-transaldolase and sulfo-transketolase pathways; and two non-glycolytic catabolic pathways, sulfo-ASMO and sulfo-ASDO, which use mono- or dioxygenases to cleave the C–S bond of SQ via alkanesulfonate monooxygenase or dioxygenase, respectively. Among these, the sulfo-Embden–Meyerhof–Parnas (sulfo-EMP) pathway is best characterized and serves as a model for SQ degradation [5], particularly in Escherichia coli [6]. ”

Comments 2: The authors found a short form (yihTUVW) of the yih cassette in some E. coli strains, in comparison with the full form (yihOPQRSTUVW). Was the deleted part such as yihOPRQRS found at the other location or dispensed in the genome of these strains like Nissle 1917? Did the authors search the genomes and find the other SQ degradation pathways in these trains hosting the short form of yih cassette?

Response 2: In all analyzed E. coli strains carrying the short form of the yih-cassette (yihTUVW), no homologs of the other yih-cassette genes (yihO, yihP, yihQ, yihR, yihS) were detected elsewhere in the genome. We have now specified this in the Results 2.1 section. As for alternative SQ catabolic genes and pathways in the studied E. coli Nissle 1917, we actually did search for homologs of bacterial genes encoding any SQ degradation-related proteins in its genome and did not observe any promising candidates. This is now clarified in the Results section 2.2: “To assess whether the E. coli Nissle 1917 strain used other genes to utilize SQ, we identified 25 DEGs whose protein products were similar to known bacterial SQ catabolism enzymes. However, these candidates were never co-localized, thus not forming operons or cassettes, and the sequence identity between respective proteins was never higher than 35% suggesting a metabolic function other than utilization of SQ (Suppl. Table S2)”. The respective details are provided in Suppl. Table S2, which is referenced in this paragraph.

Comments 3: In EMSA experiment, the purified recombinant proteins of YihW derived from both E. coli strains should be presented as in the from of SDS-PAGE.

Response 3: The respective Supplement Figure (Suppl. Fig. 6) has now been added and referenced in the Materials and Methods section “YihW expression and purification”.

Comments 4: Further, the YihW proteins were purified through denaturing/refolding. Did you obtain the soluble or active form actually? The protein/DNA complex is somehow unusual in the native gels, which did not perform mobility. Besides, two DNA bands were observed in the EMSA gel.

Response 4: We thank the reviewer for their careful assessment and for raising important points regarding the EMSA experiments and protein purification strategy.

Indeed, the YihW-DNA complex looks non-canonical for a local regulator. We were also surprised with the type of the YihW:DNA complex first obtained by Shimada and colleagues (Shimada et al, 2019, Microbiology (Reading) 2019, 165, 78–89, 542, doi:10.1099/mic.0.000740) and YihW (CsqR) molar excess required for complete binding of the target DNA (up to 240x). This type of binding is more typical for nucleoid proteins like Dps or H-NS (Azam TA, Ishihama A. Twelve species of the nucleoid-associated protein from Escherichia coli. Sequence recognition specificity and DNA binding affinity. J Biol Chem. 1999 Nov 12;274(46):33105-13. doi: 10.1074/jbc.274.46.33105). However, in our work we obtained the same result, but with much lower molar excess required for complete binding of the target DNA. This means that our protein was active. Moreover, in case of transcriptional regulators, if they are not refolded successfully, they simply aggregate. The 8x-16x molar protein excess that we obtained in our study is typical for a local regulator, but the mode of binding is not. We suppose that this might reflect the oligomeric structure of the protein that is not yet known, and/or its properties similar to nucleoid proteins. We have not discussed the mode of binding separately because the same result had been obtained earlier by the other group.

The presence of two DNA bands on the native gels has also been observed in previous studies (e.g., Shimada et al.) and in other unrelated experiments in our lab. This phenomenon likely reflects conformational heterogeneity in the DNA substrate and does not interfere with protein–DNA complex formation or the interpretation of results.

Comments 5: Anyway, the EMSA experiments results should be further confirmed.

The binding assay was performed at two temperatures. Did these two E. coli strains show different response to SQ at various temperatures?

Response 5: We agree that EMSA results should ideally be validated with in vivo assays. We have now softened some of the conclusions in the discussion, and are currently planning to perform ChIP-seq and RNA-seq experiments for both E. coli strains under different temperature conditions to further investigate CsqR binding and regulatory activity. However, these experiments are beyond the scope of the present study and will be pursued as part of future work.

Comments 6: Finally, reference citation in the text should be kept in the order and some supplementary files such as Fig. S2 and Fig. S4 were not cited in the text. The RNA-seq data should be provided available.

Response 6: We thank the reviewer for noticing this. The citation order has been checked and corrected throughout the manuscript. Figure S2 has now been added and properly cited in the text, and references to Figures S4 and S5 have been revised and clearly indicated in the relevant sections.

The RNA-seq data generated in this study have now been deposited in the NCBI Sequence Read Archive under BioProject accession PRJNA1338229. We have added this information in “Data Availability Statement” at the end of the manuscript and in the “RNA library preparation and sequencing” subsection of the Methods.

Round 2

Reviewer 1 Report

Comments and Suggestions for Authors

The authors have thoroughly responded to all the comments and have significantly improved the quality of the manuscript.

Author Response

Comments 1:  The authors have thoroughly responded to all the comments and have significantly improved the quality of the manuscript.

Response 1: We thank the reviewer for helpful comments and suggestions.

Reviewer 2 Report

Comments and Suggestions for Authors

This version has been improved greatly, and the responses to the comments sound good. However, citation order of the literatures should be rearranged. For example, the citation references [34-39] in the text should not be before the references [13-33]. 

Author Response

Comments 1: This version has been improved greatly, and the responses to the comments sound good. However, citation order of the literatures should be rearranged. For example, the citation references [34-39] in the text should not be before the references [13-33]. 

Repsonse 1: Thank you for the comment. We have revised the Reference section and corrected the order of citations.